# Hemolysin Co-Regulatory Protein 1 Enhances the Virulence of Clinically Isolated *Escherichia coli* in KM Mice by Increasing Inflammation and Inducing Pyroptosis

**DOI:** 10.3390/toxins15030171

**Published:** 2023-02-22

**Authors:** Hao Wang, Long-Bao Lv, Li-Ping Chen, Jin-Long Xiao, Jue Shen, Bin Gao, Jin-Gang Zhao, Dong-Mei Han, Bin-Xun Chen, Shuai Wang, Gen Liu, Ai-Guo Xin, Peng Xiao, Hong Gao

**Affiliations:** 1College of Food Science and Technology, Yunnan Agricultural University, Kunming 650201, China; 2National Resource Center for Non-Human Primates, National Research Facility for Phenotypic & Genetic Analysis of Model Animals (Primate Facility), Kunming Institute of Zoology, Chinese Academy of Sciences, Kunming 650107, China; 3College of Foreign Languages, Yunnan Agricultural University, Kunming 650201, China; 4College of Veterinary Medicine, Yunnan Agricultural University, Kunming 650201, China; 5National Foot-and-Mouth Disease Para-Reference Laboratory (Kunming), Yunnan Animal Science and Veterinary Institute, Kunming 650224, China

**Keywords:** *E. coli*, Hcp1, pathogenicity, pyroptosis, inflammation, AKI, ALI, CRISPR/Cas9

## Abstract

Hemolysin-coregulated protein 1 (Hcp1) is an effector released by the type VI secretion system (T6SS) in certain pathogenic strains of *Escherichia coli* (*E. coli*) that causes apoptosis and contributes to the development of meningitis. The exact toxic consequences of Hcp1 and whether it intensifies the inflammatory response by triggering pyroptosis are yet unknown. Here, utilizing the CRISPR/Cas9 genome editing method, we removed the gene expressing Hcp1 from wild-type *E. coli* W24 and examined the impact of Hcp1 on *E. coli* virulence in Kunming (KM) mice. It was found that Hcp1-sufficient *E. coli* was more lethal, exacerbating acute liver injury (ALI) and acute kidney injury (AKI) or even systemic infections, structural organ damage, and inflammatory factor infiltration. These symptoms were alleviated in mice infected with W24Δ*hcp1*. Additionally, we investigated the molecular mechanism by which Hcp1 worsens AKI and found that pyroptosis is involved, manifested as DNA breaks in many renal tubular epithelial cells. Genes or proteins closely related to pyroptosis are abundantly expressed in the kidney. Most importantly, Hcp1 promotes the activation of the NLRP3 inflammasome and the expression of active caspase-1, thereby cleaving GSDMD-N and accelerating the release of active IL-1β and ultimately leading to pyroptosis. In conclusion, Hcp1 enhances the virulence of *E. coli*, aggravates ALI and AKI, and promotes the inflammatory response; moreover, Hcp1-induced pyroptosis is one of the molecular mechanisms of AKI.

## 1. Introduction

It has been revealed that certain Gram-negative bacteria have a complex secretion mechanism called the type VI secretion system (T6SS) [1,2,3]. Its effectors are cytotoxic, as shown in previous studies [4,5,6]. Hemolysin-coregulated protein 1 (Hcp1) is a member of the Hcp protein family found in *Escherichia coli* (*E. coli*), which is unique to T6SS and makes up the bulk of its extracellular structure [7,8]. Hcp1 contributes to the virulence of *Edwardsiella ictaluri* in catfish [9]. In *Burkholderia pseudomallei*, Hcp1 stimulates strong IFN-γ-secreting T-cell responses in patients with acute melioidosis and correlates with patient survival [10]. Additionally, in *E. coli* RS218, Hcp1 is injected into the host cell HBMEC as a bacterial toxin via the T6SS to promote cysteinyl aspartate specific protease-8 (caspase-8) expression and induce apoptosis [11]. Similarly, Zhao et al. found that Hcp1 can promote apoptosis and cause *E. coli* meningitis [12].

Pyroptosis is a new form of cell death, which is defined as gasdermin-mediated programmed death [13,14,15]. The classical pathway of pyroptosis is mediated by caspase-1 [16]. In the classical pathway of pyroptosis, NOD-like receptor thermal protein domain-associated protein 3 (NLRP3) inflammasome assembly initiates the proteolytic cleavage of dormant pro-caspase-1 into active caspase-1, which transforms the cytokine precursors pro-IL-1β and pro-IL-18 into mature and biologically active interleukin-1β (IL-1β) and interleukin-18 (IL-18) [14,17,18,19]. In addition, lipopolysaccharide (LPS) is a typical bacterial toxin protein that can activate caspase-11 to induce pyroptosis [20]. Wanyan Deng showed that SpeB, a bacterial toxin secreted by group A Streptococcus, can target the cleavage of gasdermin A (GSDMA) and induce pyroptosis [21]. Recent studies have shown that *Citrobacter freundii* T6SS activates the NLRP3 inflammasome and induces pyroptosis [22]. Although Hcp1 has been demonstrated to induce apoptosis and promote infection and meningitis [7,11,12], the exact toxic consequences of Hcp1 and whether it exacerbates the inflammatory response by triggering pyroptosis are unknown.

This work aims for a better understanding of how Hcp1 contributes to the virulence of *E. coli*. We initially removed Hcp1 from wild-type *E. coli* W24 (WT W24) with CRISPR/Cas9, then evaluated the virulence and histopathological damage posed by WT W24 and Hcp1 knockout *E. coli* (W24Δ*hcp1*) strains and studied the molecular mechanism of how Hcp1 induces pyroptosis and promotes the occurrence of acute kidney injury (AKI).

## 2. Results

### 2.1. Deletion of E. coli Toxin Gene Hcp1

To investigate the role of Hcp1 in *E. coli* pathogenesis, we deleted the gene expressing Hcp1 in WT W24. The gene structure of Hcp1 is shown in Figure 1A. We designed two sgRNAs to target the Hcp1 gene. Gene editing is shown in Figure 1B. Finally, W24Δ*hcp1* was obtained, which was confirmed by PCR and gene sequencing (Figure 1C,D).

### 2.2. Hcp1 Enhances E. coli Virulence against KM Mice

First, we established a mouse model of acute infection caused by *E. coli* (Figure 2A). Dissection revealed that Hcp1-sufficient WT W24 induced severe organ damage, a consequence similar to that caused by direct injection of LPS. In contrast, in mice infected with the W24Δ*hcp1* strain, the damage was not present (Figure 2B). Acutely infected mice were clinically scored, with those infected with WT W24 showing severe damage, those injected with LPS showing moderate damage, and those infected with W24Δ*hcp1* showing mild damage. Moreover, we measured the body temperature of infected mice, which peaked 24 or 36 h after infection (Figure 2C,D). Mice infected with WT W24 developed a more serious systemic infection than those infected with W24Δ*hcp1*, with significantly increased bacterial loads in the blood, spleen, liver, and kidney. As expected, mice injected with LPS also showed serious systemic disease (Figure 2E–I).

Moreover, we found a large number of inflammatory factors, including IL-18, IL-1β, TNF-α, and IFN-γ in the serum of infected mice, and the level of inflammatory factors in the serum of mice infected with the W24Δ*hcp1* strain was reduced in the serum of mice infected with WT W24. (Figure 2J). Additionally, acute infection with the WT W24 strain caused tissue damage to the lung, spleen, and small intestine, but in the mice infected with the W24Δ*hcp1* strain, these symptoms were not present (Figure 2K). 

### 2.3. Hcp1 Promotes E. coli-Induced Acute Liver Injury in Mice

WT W24 infection caused more severe liver damage than W24Δ*hcp1* infection, as revealed by higher serum ALT and AST levels in WT W24-infected mice (Figure 3A,B). The macroscopic image of the liver is presented in Figure 3C. It can be seen that the infection with Hcp1-sufficient WT W24 significantly promotes hepatomegaly. Transcript levels of IL-18 and IL-1β were significantly increased in liver tissues after WT W24 infection, and this consequence was similar to that caused by LPS injection. However, the inflammatory factor levels were significantly lower in mice with W24Δ*hcp1* infection than in those with WT W24 infection (Figure 3D), which was confirmed by liver ELISA and liver IHC (Figure 3E–G).

Given the apparent detrimental effects of Hcp1 on liver function, a histopathological examination of morphological changes in the liver was performed. Histopathology revealed that WT W24 infection caused severe congestion, hemorrhage, and structural damage in liver tissues (Figure 3F). Similarly, TEM showed that WT W24 infection resulted in chromatin alteration and liver cell death, which were alleviated in livers infected with W24Δ*hcp1* (Figure 3G). These results suggest that Hcp1 of *E. coli* promotes acute liver injury (ALI).

### 2.4. Hcp1 Promotes E. coli-Induced Acute Kidney Injury in Mice

We addressed the toxicological effects of Hcp1 on the kidney after finding that it can cause ALI. Kidneys were significantly enlarged after WT W24 infection, while the kidney index was relatively lower after W24Δ*hcp1* infection (Figure 4A). Based on serum renal function (BUN and SCr) indicators, it was found that W24 infection-induced AKI was similar to that caused by direct LPS injection in mice. In addition, WT W24 infection caused more severe renal injury than W24Δ*hcp1* infection (Figure 4B,C). IHC showed that in WT W24-infected kidneys, a large number of IL-18 and IL-1β were expressed, mainly in renal tubular epithelial cells, while their expression was reduced in W24Δ*hcp1*-infected kidneys (Figure 4D). Similarly, ELISA of kidney tissue showed that in WT W24-infected kidneys, a large number of IL-18, IL-1β, and TNF-α, were expressed, but their expression was significantly reduced in W24Δ*hcp1*-infected kidneys (Figure 4E).

Moreover, renal histopathology demonstrated that WT W24 was more virulent, manifested as the loss of more nuclei and invasion of more inflammatory cells. In extravasated blood, the glomerulus was enlarged, the renal tubular epithelial cells were necrotic, and the lumen was filled with serous fluid, fibrin, and neutrophils. No visible lesions were found in the control group. Similarly, TEM observation showed that cell damage was severe after WT W24 infection in mice. For example, nucleus pycnosis of most renal tubular epithelial cells and dissolution of small nucleoli occurred; euchromatin edge set part of the cell nucleolus side together. The microvilli on the cell surface were sparsely arranged, the base was swollen, and the pleura disappeared. Mitochondria increased and swelled. The damage was alleviated in W24Δ*hcp1*-infected livers (Figure 4D,F). These results suggest that the Hcp1 promotes AKI. After finding that Hcp1 can promote ALI, we further explored the toxicological effects of Hcp1 on the kidney.

### 2.5. Hcp1 Induces Pyroptosis in the Kidney

One of the characteristics of pyroptosis is the large production of IL-18 and IL-1β [23,24]. According to the above results, it is speculated that pyroptosis occurs in the kidneys of mice infected with *E. coli* W24. IHC results showed that each factor in the pyroptosis classical pathway NLRP3-caspase-1-GSDMD was mainly expressed in renal tubular epithelial cells. Furthermore, WT W24 infection more significantly promotes the expression of these factors than W24Δ*hcp1* infection (Figure 5B–E). Additionally, mRNA levels of pyroptosis-related genes (NLRP3, ASC, caspase-1, GSDMD, IL-18, and IL-1β) in the kidneys with W24Δ*hcp1* infection are considerably lower than in those with WT W24 infection (Figure 5A).

Genomic DNA in pyroptosis nuclei is disrupted, exposing 3′-OH and making the cells TUNEL positive [16,25]. WT W24 infection can promote DNA fragmentation in renal tubular epithelial cells, increasing the TUNEL-positive ratio; in W24Δ*hcp1*-infected kidneys, TUNEL-positive cells are significantly fewer (Figure 6A,B). Furthermore, WB results showed that WT W24 infection activated the NLRP3 inflammasome, while the degree of inflammasome activation was significantly reduced in W24Δ*hcp1*-infected kidneys, which was consistent with the IHC results. In addition, we also found that caspase-1 was activated in infected kidneys, which was facilitated by Hcp1-replete *E. coli* (Figure 6C,D). We observed the cleavage of GSDMD in WT W24-infected kidneys and found abundant GSDMD-N, an executor of pyroptosis [26,27]. Furthermore, active forms of IL-1β were found in infected kidneys, and they were significantly reduced in W24Δ*hcp1*-infected kidneys (Figure 6C,D). These results validate our hypothesis that pyroptosis occurs in W24 *E. coli* infection-induced ALI and that Hcp1-sufficient W24 promotes pyroptosis and then exacerbates ALI.

### 2.6. Hcp1 Binds to NLRP3 Protein

Section 2.5 show that Hcp1 can promote NLRP3 expression. It is hypothesized that Hcp1 will be detected by NLRP3 after entering cells and might adhere to it, promoting the expression of NLRP3. HDOCK SERVER was used for protein docking, and then SWISS-MODEL was used to generate the three-dimensional structure of Hcp1. This structure is displayed in Figure 7A (PDB: 4W64). Then, AlphaFold was used to obtain the 3D NLRP3 result (Figure 7B).

HDOCK SERVER predicted 10 good combination models (docking scores and confidence scores are shown in Appendix A). The optimal results we selected (Figure 7C) show that Hcp1 can combine well with NLRP3, with an interconnection score of −266.70 and a confidence score of 0.9117 (the docking score is around −200 and when the confidence score is above 0.7, it is better, and the two molecules are most likely to bind), which supports our hypothesis.

## 3. Discussion

Our results demonstrate that Hcp1 can increase the virulence of *E. coli* in KM mice, leading to a more serious systemic infection and promoting both ALI and AKI. We found that pyroptosis is involved in AKI and is highly correlated with Hcp1, and Hcp1 promotes the expression of pyroptosis-related genes or proteins in the kidney. Furthermore, pyroptosis promotes the activation of the NLRP3 inflammasome and promotes the expression of active caspase-1, which cleaves GSDMD-N and accelerates the release of active IL-1β and ultimately leads to cell death. These studies reveal that Hcp1 is an essential bacterial toxin in pathogenic *E. coli* and promotes AKI by inducing pyroptosis.

In this study, we identified and isolated WT W24 (Appendix A) from the feces of diarrheal piglets from a pig farm in Yunnan, China, and performed a sequencing analysis, which has not been published yet. Hcp1, a bacterial-derived toxin protein of the T6SS, can also be secreted into host cells to cause damage [28,29]. When mice were infected with WT W24, we found high levels of TNF-α, IFN-γ, IL-1β, and IL-18 in their blood, livers, and kidneys. These are typical inflammatory factors, and their massive release means a robust inflammatory response e [30]. In WT W24, Hcp1 increases bacterial load across tissues and promotes systemic infection; *E. coli* PCN003 infection achieves similar results [7]. In *E. coli* RS218, Hcp1 causes the release of massive IL-6 and IL-8. The induction of HBMC apoptosis by promoting the expression of active caspase-8 is a mechanism for inducing bacterial meningitis [11]. These results suggested that Hcp1 could promote inflammation and enhance the pathogenicity of different strains. Although Hcp1 has been shown to promote injury as a one-minute toxin protein, no systematic studies on its pathogenicity have been conducted. Our study found that Hcp1-expressing *E. coli* W24 was more lethal, promoting systemic infection like strain PCN003. We also observed pathological damage to internal organs, including the spleen, lung, and small intestine, but the damage was alleviated in W24Δ*hcp1*-infected mice. These findings imply that Hcp1 increases the pathogenicity of *E. coli* in KM mice.

We also found that the damage to the liver caused by *E. coli* W24 is typical ALI [31], which is similar to that induced by LPS. WT W24 damages liver function and structure and promotes the expression and release of many inflammatory factors. Furthermore, it can also be observed by TEM that liver cells are in a state of death edge, and all of these phenomena are highly correlated with Hcp1. Our data also show that *E. coli* W24 promotes AKI, which is consistent with the findings in previous studies [32,33]. The attenuation of kidney and liver function, the infiltration of inflammatory cells, the expression and release of inflammatory factors, and the accumulation of a large number of mitochondria in renal tubular epithelial cells were observed by TEM. The reason may be that the host needs a lot of energy and ROS to clear bacteria after infection, and progress has been made in studies on this topic. In contrast, these problems were alleviated in W24Δ*hcp1*-infected mice, suggesting that Hcp1 can act as a toxin protein to enhance the virulence of *E. coli*, causing severe damage in mice.

Pyroptosis is a gene-regulated programmed death whose canonical pathway is mediated by the NLRP3 inflammasome [34,35]. Recent studies have found that Citrobacter freundii can activate the NLRP3 inflammasome through the T6SS and cause pyroptosis in BMMs. Furthermore, one of the hallmarks of pyroptosis is the massive release of IL-18 and IL-1β [23,24]. Our results show thatin *E. coli*-induced AKI, Hcp1 can promote the massive expression and release of IL-18 and IL-1β in the kidney. It is speculated that pyroptosis is involved in *E. coli*-induced AKI, and Hcp1 can promote pyroptosis. The research findings are in line with the expectations. Hcp1 promotes the activation of the NLRP3 inflammasome and the production of active caspase-1. Active caspase-1 cleaves GSDMD, and pro-IL-1β generates GSDMD-N to induce pyroptosis, this is a typical pyroptosis pathway [36]. After WT W24 strain infection, NLRP3, caspase-1, GSDMD-N, and other factors in the classical pyroptosis pathway were found in the kidney, mainly in renal tubular epithelial cells. Moreover, TUNEL staining shows that many renal tubular epithelial cells have DNA breaks, which is another sign of pyroptosis [16,25]. DNA breaks were localized in renal tubular epithelial cells, suggesting that pyroptosis may occur in there, but further studies are needed. Although our findings indicate that pyroptosis is weakened following Hcp1 infection, it can still be found in the Hcp1 infection group. The results suggest that other virulence factors in *E. coli* may promote pyroptosis, which needs further confirmation. Additionally, using protein docking, we found that Hcp1 and NLRP3 would bind together in idealized situations. The result requires additional experimental confirmation using techniques such as CO-IP. Evidence suggests that pyroptosis is one cause of AKI [37]. Hcp1 can promote the occurrence of AKI in mice and the pyroptosis of renal tubular epithelial cells. It is suggested that Hcp1-induced pyroptosis is one of the mechanisms by which *E. coli* causes AKI.

This study has certain limitations. Although it is confirmed that pyroptosis is involved in AKI and pyroptosis-related proteins are mainly expressed in renal tubular epithelial cells, cell experiments still need to be conducted for further research. In addition, our research is limited to the classical signaling pathway of pyroptosis. Whether the non-classical pathway is involved in Hcp1-promoted AKI requires further investigation. 

In conclusion, Hcp1 can increase the pathogenicity of WT W24, promotes ALI and AKI in KM mice, and accelerates inflammatory responses. Hcp1-induced pyroptosis is one of the molecular mechanisms of AKI.These results provide new insights into the pathogenic mechanism of pathogenic *E. coli*, and revealed the virulence contribution of Hcp1 as an essential bacterial toxin in pathogenic *E. coli.*

## 4. Materials and Methods

### 4.1. Strains and Plasmids

All *E. coli* W24 and *E. coli* DH5α strains were grown at 37 °C or 30 °C in Luria-Bertani (LB) broth with appropriate antibiotics and shaking. The bacterial strains and plasmids used in this study are given in Appendix A.

### 4.2. Deletion of E. coli Hcp1

In brief, for the convenience of gene editing, we used the primers in Appendix A to construct the pUC-spacer-Donor plasmid according to the process in Appendix A.

W24 competent cells harboring pCas were prepared as described previously [38]. In accordance with the procedure, arabinose (30 mM final concentration) was given to the culture for λ-Red induction. The transformation was done by electroporation using the GenePulser Xcell Electroporation System (Bio-Rad, Hercules, CA, USA). For electroporation, 80 µL of cells were mixed with 200 ng of pUC-spacer-Donor series DNA; electroporation was done in a 2 mm Gene Pulser cuvette (Bio-Rad) at 1.8 kV, and the product was suspended immediately in 1 mL of ice-cold SOC medium. Cells were recovered at 30 °C for 2 h before being spread onto LB agar containing kanamycin (50 mg/L) and Ampicillin (100 mg/L) and incubated overnight at 30 °C. Transformants were identified by colony PCR, and DNA sequencing was prepared as described previously [39,40]. 

### 4.3. KM Mice Infection Model

The Experimental Animal Center at Kunming Medical University sold us Kunming mice (20–25 g). (Kunming, China). Mice were randomly allocated to experimental animal groups and kept in a pathogen-free cage, in a room with a photoperiod of 12 h of light and 12 h of darkness, at a temperature of 25 ± 2 °C and ambient humidity levels of 40–60%. The Yunnan Agriculture University’s Animal Ethics Committee (Kunming, China) approved the experimental program.

The mice intraperitoneally injected with LPS were set as positive controls since it has been demonstrated that LPS may cause inflammation to accelerate pyroptosis [20,36]. Sixty Kunming mice were randomized into four groups: the control group, the WT W24 group, the W24*Δhcp1* group, and the LPS group. The LPS group was only intraperitoneal (i.p.) injected with LPS (4 mg/kg, mice), the control group was i.p. injected with LB, the WT W24 group and the W24*Δhcp1* group were challenged with 2 mL of *E. coli* strains (1 × 10^7^ CFU/mL) by an i.p. injection to evaluate the pathogenicity of the different *E. coli* strains. The survival rate and body temperature were recorded.

### 4.4. Clinical Signs and Assigned Clinical Scores

Following Figure 3A, all experimental KM mice were scored and assigned clinical scores as follows [41]: 0 indicates a normal reaction to stimuli, 1 a ruffled coat and a delayed response, 2 a response to repeated stimuli only, 3 an inability to respond or circling in place, and 4 a dead. Mice were mercifully put to death if they had significant lethargy or neurological signs (score = 3).

### 4.5. Biochemical Analysis and ELISA Assay

A tail-vein bleed collected blood from mice 24 h after infection. An automated biochemical analyzer model 7020 was used to test the levels of AST, ALT, BUN, and SCr (Hitachi, Chiyoda-ku, Japan).

The levels of IL-1β, IL-18, TNF-α and IFN-γ (Jiancheng Bioengineering, Nanjing, China) in serum or tissue samples were determined by ELISA kit, according to the manufacturer’s protocols.

### 4.6. Liver and Kidney Organ Index

36 h after infection, the mice in each group were weighed (free access to water), the liver and kidneys were removed in a clean environment, the surface fat and mesangium were removed, and an electronic balance was used to weigh and record the results. The organ index was calculated using the formula: organ index = organ weight (mg)/body weight (g).

### 4.7. Histopathology

Tissues were fixed with 4% paraformaldehyde overnight at 4 °C, followed by dehydration through an alcohol-xylene series, and finally embedded in paraffin. Leica 2235 (Wetzlar, Germany)was used to cut the kidney sections (5 m thick), which were then dried at 37 °C, deparaffinized, rehydrated utilizing a succession of xylene-alcohol solutions, washed with deionized water, and lastly stained with H&E (Servicebio, Wuhan, China).

### 4.8. Immunohistochemistry

Antigen extraction was performed using the UltraSensitiveTM SP (Mouse/Rabbit) IHC Kit (MXB, Fuzhou, China). The procedures are as follows. The slides were immersed in 0.01 M citrate buffer (pH 6.0) for 10 min in 3% H_2_O_2_. After that, the non-immune goat serum was blocked for 10 min and then incubated at 4 °C for 12 h with the inducing agents IL-1β (Abs,126104), IL-18 (Absin, 125418), NLRP3 (Absin, 14F468), caspase-1 (Santa Cruz, 14F468), and GSDMD (Abcam, ab219800. Then, the slices were washed 3 times with PBS (MXB, Fuzhou, China) buffer and placed with the biotin-labeled secondary antibody (from UltraSensitiveTM SP (Mouse/Rabbit) IHC Kit) at room temperature for 30 min. Subsequently, the slides were stained with DAB (MXB, Fuzhou, China) for 5 min and then with hematoxylin for 10 min. The images were captured using an Olympus BX43F microscope (Tokyo, Japan). The IL-1β, IL-18, NLRP3, caspase-1, and GSDMD mean densities were measured using ImageJ 1.8.0.

### 4.9. Transmission Electron Microscopy

Two percent glutaraldehyde was used to fix renal tissue samples. Then, they were washed in PBS (0.01 M) and postfixed with 1% osmium tetroxide. After gradient dehydration in acetone, the tissue samples were embedded in Araldite M (Sigma Aldrich, St. Louis, MO, USA). Ultrathin sections were cut using an ultramicrotome (Leica, Wetzlar, Germany) and stained with uranyl acetate and lead citrate. The sections were examined with a transmission electron microscope (H-7700, Hitachi, Japan).

### 4.10. RNA Extraction and Real-Time PCR

Following the manufacturer’s instructions, total RNA was extracted from mouse tissues using a Takara-made RNAiso Plus kit (Dalian, China). The PrimeScript RT Master Mix kit was used to reverse-transcribe one microgram of total RNA into cDNA (Takara, Dalian, China). SYBR Premix Ex Taq II (Takara, Dalian, China) was used for real-time PCR experiments, and CFX96TM real-time equipment was used for analysis (Bio-Rad, Hercules, CA, USA). The housekeeping gene β-actin was the reference to which relative changes in mRNA were computed using the ΔΔCt method. Appendix A provides the primer sequences that were employed, and Appendix A provides the PCR conditions and reaction mixtures.

### 4.11. TUNEL Staining

The previously stated procedures were utilized to make kidney slices, and a Colorimetric TUNEL Apoptosis Assay Kit (Beyotime, Haimen, China) was used to look for cellular DNA damage. The processes were carried out as directed by the manufacturer.

### 4.12. Western Blotting Analysis

The procedure for Western blot analysis has been reported [19]. Briefly, kidney tissues were collected according to the experimental protocol. Then, proteins were extracted from the liver tissue using a protein extraction kit (Thermo #78510). Then, separated by SDS–PAGE, and transferred onto a polyvinylidene fluoride (PVDF) membranes. The PVDF membranes were probed with the following primary antibodies: anti-NLRP3 (#ab270449), anti-ASC (#ab180799), anti-caspase-1 (#ab179515), anti-GSDMD (#ab209845) and anti-IL-1β (ab283818); for these specific proteins, anti-β-actin (Sigma-Aldrich, St. Louis, MO, USA) was used as a loading control.

### 4.13. Protein Docking

The three-dimensional structures of Hcp1 and NLRP3 were predicted using SWISS-MODEL (SWISS-MODEL (expasy.org, accessed on 19 October 2022)) and AlphaFold (AlphaFold Protein Structure Database (ebi.ac.uk, accessed on 19 October 2022)), respectively. HDOCK SERVER (HDOCK Server (hust.edu.cn, accessed on 19 October 2022)) was used for protein docking once the two proteins’ three-dimensional structures were determined [42].

### 4.14. Statistical Analysis

The findings are given as mean ± SD and were statistically analyzed using GraphPad Prism 9.0 software (GraphPad Software, San Diego, CA, USA). Analysis of variance (ANOVA) or a *t*-test was used to analyze the differences between groups. A *p*-value of <0.05 was regarded as significant.

## Figures and Tables

**Figure 1 toxins-15-00171-f001:**
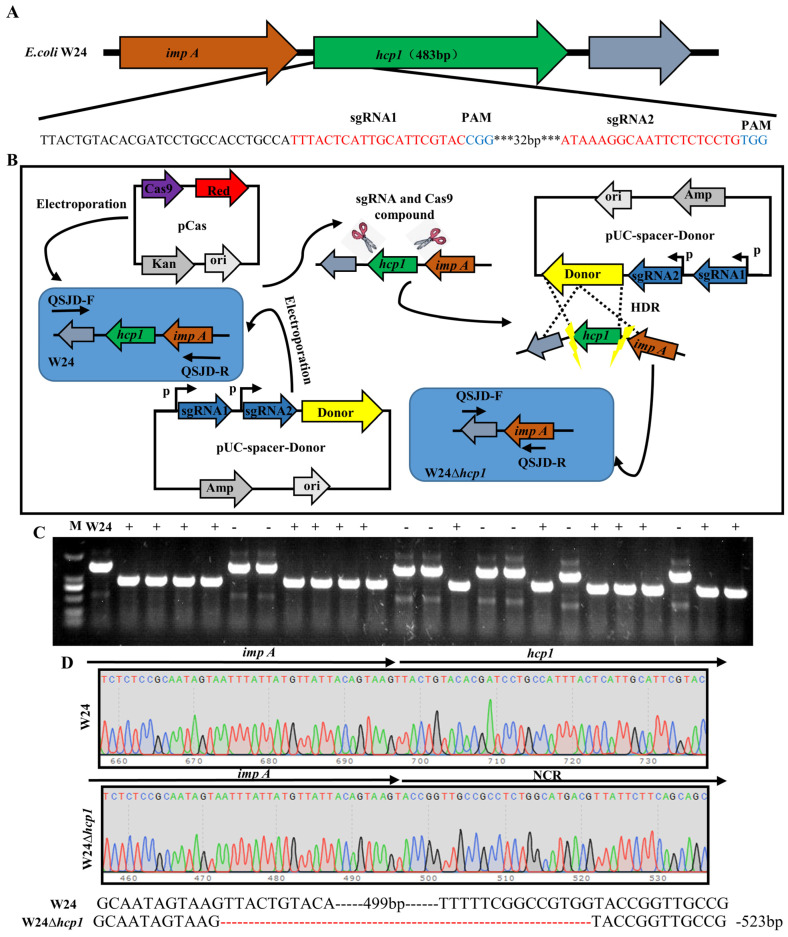
Construction and identification of *hcp1*gene knockout strain. (**A**). The gene structure of Hcp1 and the checkpoint of sgRNA (The 32bp base that was left out is denoted by the symbol ***). (**B**). *hcp1* gene knockout strategy(Homologous recombination of gene segments is shown by the dashed line). (**C**). Identification of the W24Δ*hcp1* strain by PCR. Primers QSJD-F/QSJD-R were designed to identify W24Δ*hcp1* strain; the size of the amplified bands was approximately 1455 bp or 932 bp (These numbers represent fragment sizes of PCR products from different clones, A positive clone is denoted by +, and while a negative clone is denoted by −). (**D**). *hcp1* gene KO schematic diagram and *hcp1* gene KO sequencing results.

**Figure 2 toxins-15-00171-f002:**
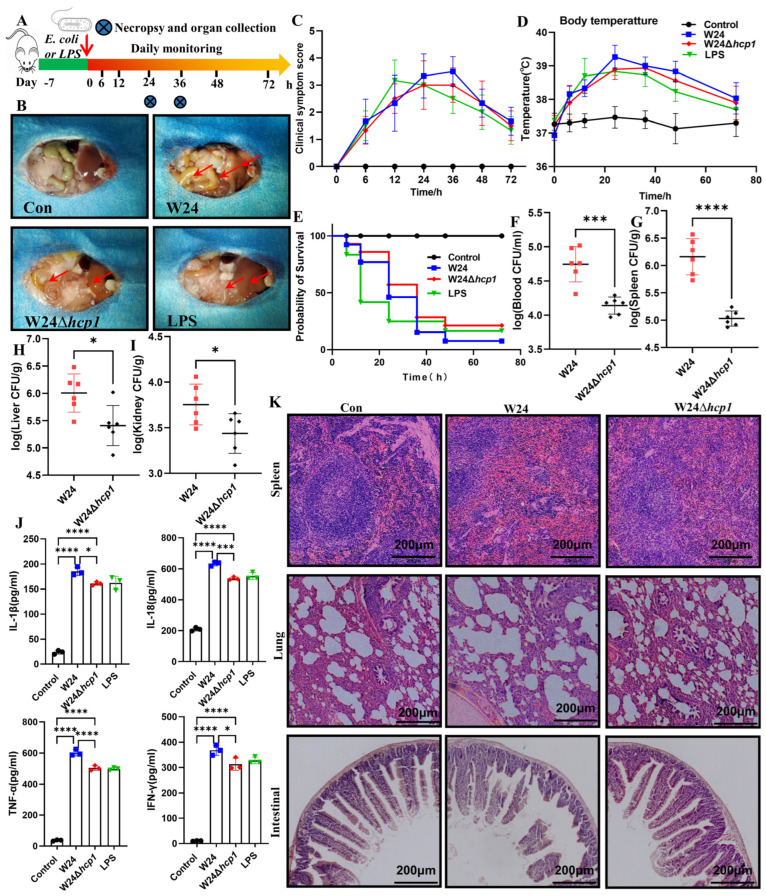
Hcp1 enhances *E. coli* virulence against Kunming (KM) mice. (**A**). Schematic diagram of KM mouse infection model. (**B**). Anatomical changes of KM mice after infection. (**C**). Clinical score of mice after infection (typical injuries are depicted by the red arrows). (**D**). Changes of body temperature in mice at different time points after infection. (**E**). Mortality of KM mice in different infection groups. (**F**–**I**). Bacterial load in different organs of mice in different infection groups (*n* = 6). (**J**). Changes of inflammatory factors in the serum of infected mice, including IL-18, IL-1β, TNF-α and IFN-γ (*n* = 3). (**K**). Pathological changes of spleen, lung and small intestine in infected mice (scale bar, 200 μm). All data are shown as the mean ± SD. **** *p <* 0.0001, *** *p <* 0.001, and * *p* < 0.05.

**Figure 3 toxins-15-00171-f003:**
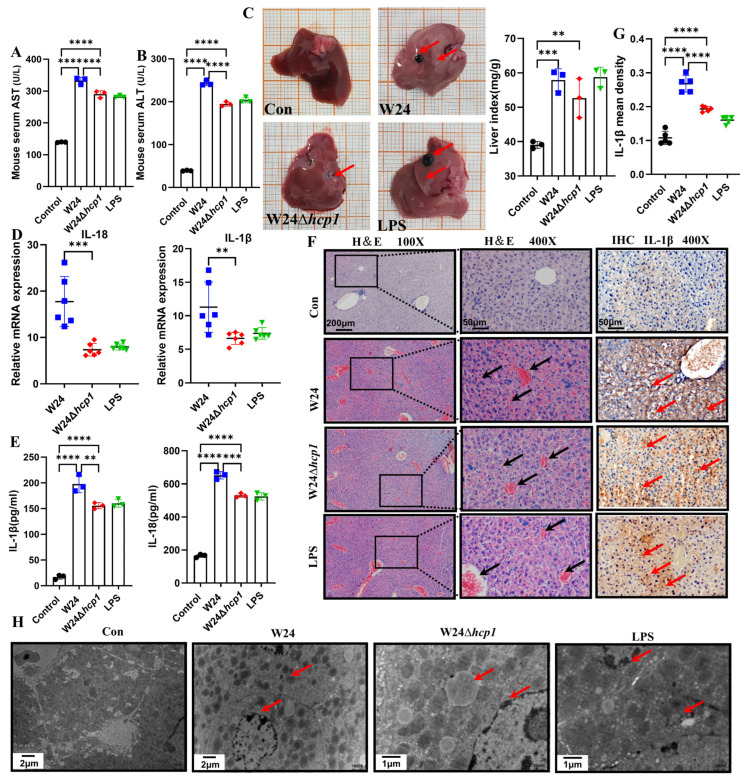
Hcp1 promotes *E. coli*-induced acute liver injury (ALI) in mice. (**A**,**B**). *E. coli* Hcp1 promote the destruction of liver function, leading to the increase of AST and ALT (*n* = 3). (**C**). The macroscopic picture of the liver and mouse liver index (*n* = 3, typical injuries are depicted by the red arrows). (**D**). The relative expression levels of IL-18 and IL-1β were determined using qRT-PCR in mice liver specimens (*n* = 6). (**E**). IL-1β and IL-18 were measured with ELISA in liver samples (*n* = 3). (**F**). Histopathological changes in acute injury liver (100×, 400×. Scale bar, 200 μm, 50 μm, typical injuries are depicted by the red or black arrows, the local zoom location is indicated by the black square box). Representative photomicrographs of immunohistochemistry results showing the staining of IL-1β in different groups. Original magnification (400×. Scale bar, 50 μm). (**G**). The mean density of IL-1β in the livers was determined (*n* = 5). (**H**). Transmission electron microscope observation of *E. coli* Hcp1 damage to hepatocytes (typical injuries are depicted by the red arrows). All data are shown as the mean ± SD. **** *p <* 0.0001, *** *p <* 0.001, and ** *p <* 0.01.

**Figure 4 toxins-15-00171-f004:**
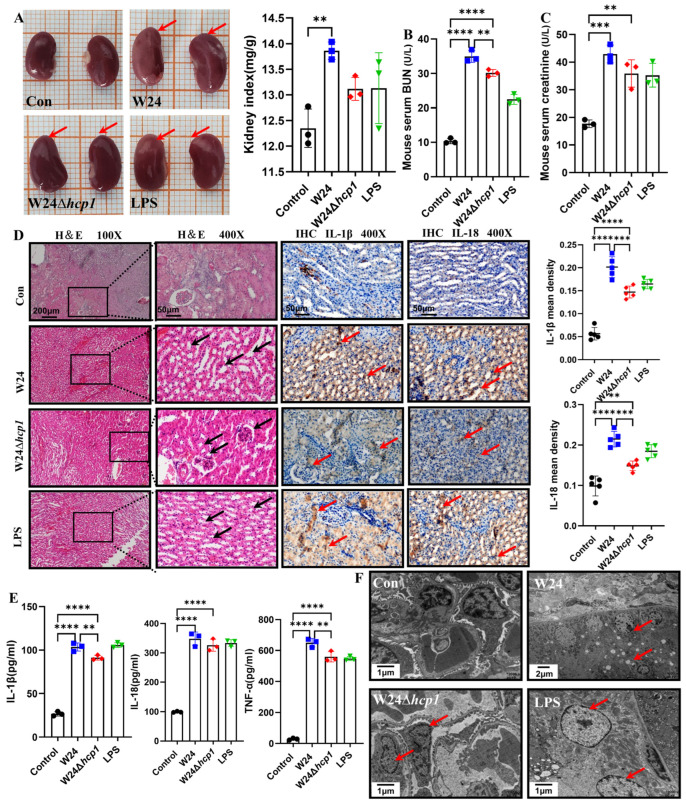
Hcp1 promotes *E. coli*-induced acute kidney injury (AKI) in mice. (**A**). The macroscopic picture of the kidney and mouse kidney index (*n* = 3, typical injuries are depicted by the red arrows). (**B**,**C**). *E. coli* Hcp1 promotes the destruction of liver function, increasing BUN and SCr (*n* = 3). (**D**). Histopathological changes in acute injury kidney (100×, 400×. Scale bar, 200 μm, 50 μm, typical injuries are depicted by the red or black arrows, the local zoom location is indicated by the black square box). Representative photomicrographs of immunohistochemistry results showing the staining of IL-18 and IL-1β in different groups. Original magnification (400×. Scale bar, 50 μm). (**E**). IL-1β, IL-18, and TNF-α were measured with ELISA in liver samples (*n* = 3). (**F**). Transmission electron microscope observation of *E. coli* Hcp1 damage to hepatocytes (typical injuries are depicted by the red arrows). All data are shown as the mean ± SD. **** *p <* 0.0001, *** *p <* 0.001, and ** *p <* 0.01.

**Figure 5 toxins-15-00171-f005:**
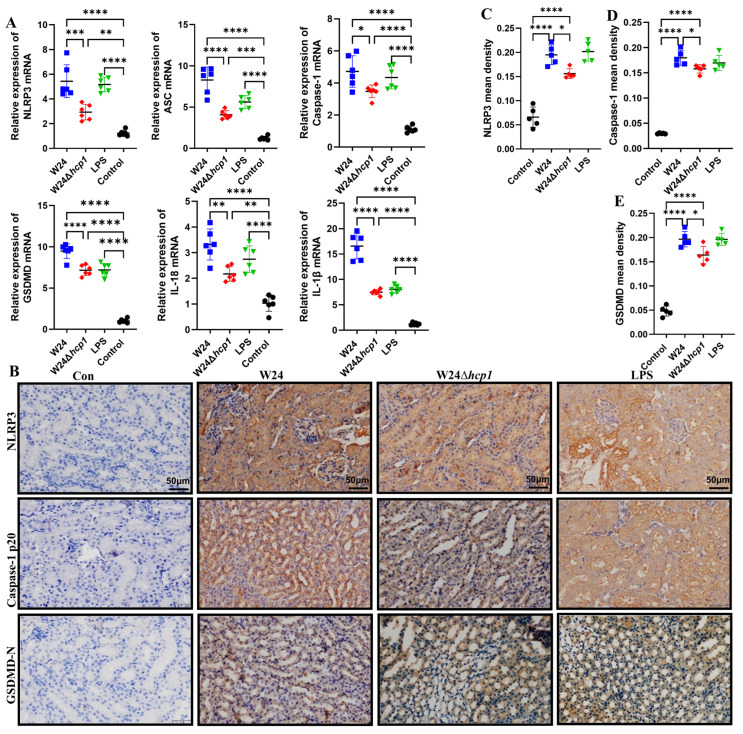
Hcp1 promotes pyroptosis in the kidney. (**A**). The relative expression levels of NLRP3, ASC, caspase-1, GSDMD, IL-1β, and IL-18 were determined using qRT-PCR in mice kidney specimens (*n* = 6). (**B**). Representative photomicrographs of immunohistochemistry results showing the staining of NLRP3, caspase-1, and GSDMD in different groups. Original magnification (400×. Scale bar, 50 μm). (**C**–**E**). The livers’ mean density of NLRP3, caspase-1, and GSDMD was determined (*n* = 5). All data are shown as the mean ± SD. **** *p <* 0.0001, *** *p <* 0.001, ** *p <* 0.01, and * *p* < 0.05.

**Figure 6 toxins-15-00171-f006:**
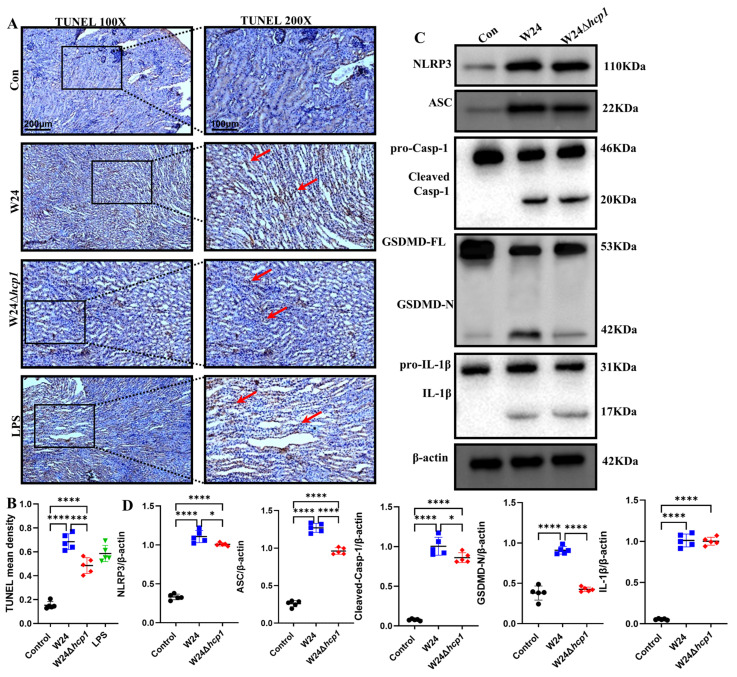
Hcp1 promotes pyroptosis. (**A**). TUNEL staining of kidneys in different treatment groups (100×, 200×. Scale bar, 200 μm, 100 μm, typical injuries are depicted by the red arrows, the local zoom location is indicated by the black square box). (**B**). Semi-quantification of TUNEL-positive cells in the kidney (*n* = 5). (**C**,**D**). Western blotting was used to detect the expression levels of NLRP3, ASC, caspase-1, GSDMD, IL-1β, and IL-18 in the kidneys of mice in different treatment groups and compared with β-actin (*n* = 5). All data are shown as the mean ± SD. **** *p <* 0.0001, *** *p <* 0.001, and * *p* < 0.05.

**Figure 7 toxins-15-00171-f007:**
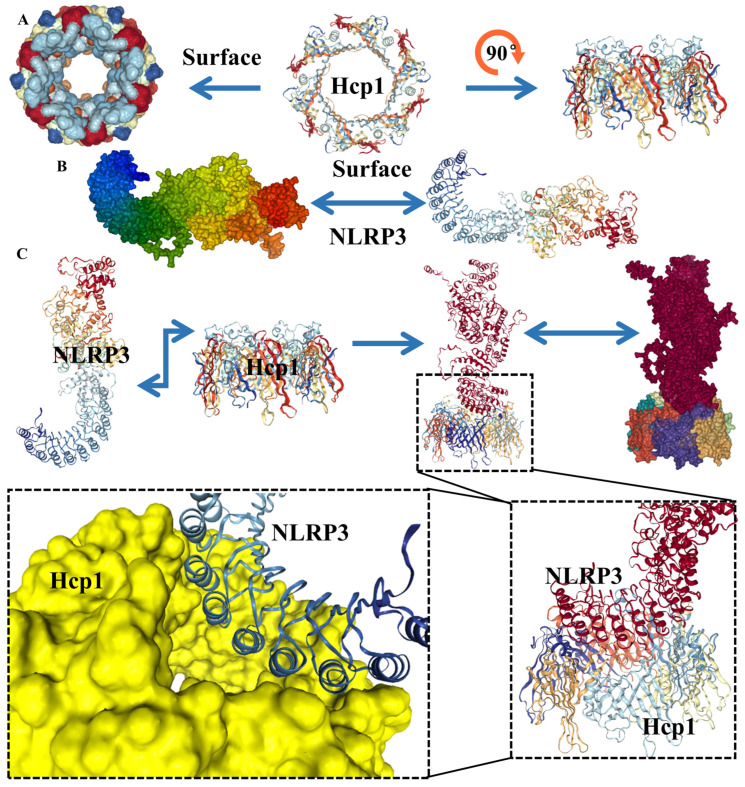
Protein structure prediction. (**A**). Three-dimensional structure of Hcp1 protein, obtained by SWISS-MODEL. (**B**). Three-dimensional structure of NLRP3 protein, obtained by AlphaFold. (**C**). The optimal binding structure of Hcp1 protein and NLRP3 protein with the local magnification of the binding site.

## Data Availability

The data presented in this study are available in this article.

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
