# Peer review of "Hemolysin Co-Regulatory Protein 1 Enhances the Virulence of Clinically Isolated Escherichia coli in KM Mice by Increasing Inflammation and Inducing Pyroptosis"

_toxins, 2023, doi:10.3390/toxins15030171_

Round 1

Reviewer 1 Report

I would like to thank the authors for their efforts to do this interested study which describing the contribution of Hcp1 protein in the virulence factors of E. coli by comparison between wild-type E. coli W24 (WT W24) with Hcp1 knockout E. coli (W24ΔHcp1) strains. So, the virulence and histopathological damage were evaluated in mice infected by WT W24 and W24ΔHcp1 strains. Also, molecular mechanism of how Hcp1 induces pyroptosis and promotes the occurrence of acute kidney injury (AKI) was investigated.

Introduction is well written and described the main aim of the word. Material and methods section was correctly designed. Experimental design was correctly arranged.

There are interested issues in the current study which led me to judge the present manuscript as “acceptable”. For example, the results showed that Hcp1 promotes the expression of pyroptosis-related genes or proteins in the mice kidney and concluded that Hcp1 is an essential bacterial toxin in E. coli and promotes AKI by inducing pyroptosis. In addition, the results revealed that Hcp1 is an essential bacterial toxin in E. coli and promotes liver damage. Several molecular mechanisms revealed the virulence contribution of Hcp1 as an essential bacterial toxin in E. coli.

Reviewer 2 Report

General comments

The topic says the E. coli used was clinical. However, in the discussion, the authors say they isolated the organisms from pigs (veterinary).

In the materials and methods, the authors should specify the source of the bacterial strains they used. Also, Supplementary Tab. 1 is not found. The same applies to Supplementary Fig 1 and all the other supplementary materials mentioned.

The authors also indicate that "Transformants were identified by colony PCR and DNA sequencing" but fail to give details of these. If these two procedures were performed according to a reference, then please provide this.

The results section is overcrowded. The authors must make it more specific and carry explanations to the discussion section. Also, there is no need to repeat the methods in each of the subsections here. This makes the results unnecessarily lengthy. See Lines 162-184 for example. Furthermore, the figures are super crowded and difficult to understand. The authors must present only salient figures and supporting images like gels and procedures should be moved to the appendix or supplementary materials.

In the discussion, first paragraph, the authors indicate that “Hcp1 is an essential bacterial toxin in E. coli”. However, the virulence mechanisms in E. coli differs by pathotype and not all the pathotypes produce toxins. Furthermore, the authors do not indicate what pathotype they worked with. Was this one of the diarrhoeagenic ones or was it an extraintestinal strain? Also, the role of Hcp protein family has been fully demonstrated in the literature and in the current study, the authors fail to indicate what new knowledge they add. Although it is important to confirm existing findings, a small degree of novelty should be introduced to provide the significance of the study

Furthermore, in most parts of the discussion, the authors repeat their results in different words, instead of discussing the findings. This makes the discussion limited. They authors must discuss their results, comparing them with existing findings and providing implications of the study. What is the overall value of the study? How useful are the findings?

Specific comments

Lines 310-314: This whole section must be written again. It is poorly written with some sentences not making sense and others sounding like instructions.

Line 326: What is the role of "Jiancheng Bioengineering, Nanjing, China?" Is that where the tests were performed?

Line 332: an electronic balance

Line 342-343: Rephrase.

Line 348: Which secondary antibody?

Line 354: renal tissue samples

Line 365: Delete "piece of"

Section 4.10: Provide the PCR conditions and reaction mixtures. Indicate the use of a no template control.

Line 374: Rephrase the starting

Line 376: Which primary antibodies?

Line 378: Which findings? Protein structure? DNA damage?

Line 87-88: Provide p-value

Line 97-98: This is meant for discussion

Reviewer 3 Report

The manuscript entitled "Hemolysin co-regulatory protein 1 enhanced the virulence of clinically isolated Escherichia coli in KM mice by increasing inflammation and inducing pyroptosis" submitted by HONG gao et al. describes the Hcp1 protein shows increased virulence of E. coli in Kunming Mouse (KM) by inducing pyroptosis. Please see comments below.

1) The authors need to mention the abbreviations such as LPS, GSDMA, NLRP3 and all the abbreviations must be elaborated when they first appear in the manuscript. A reader can not follow if the authors mention the abbreviations without elaborating them. For example, line 44, LPS (what it is?, please elaborate it). Similarly, for other words throughout the manuscript.

2) It is not clear from the study design that why LPS was chosen for the study? Please provide a rationale.

3) In figure 5a, please include controls besides W24, W24 HCP1 and LPS. Discuss the results of 5a in the manuscript as well.

4) In the title, the authors have mentioned that Hcp1 causes increased inflammation did not see any of this inflammation in the discussion section. Please provide reasonable rationale how this was observed.

Please do not accept until the necessary revisions are made.

Round 2

Reviewer 2 Report

Thank you for addressing the comments raised

Author Response

Thank you very much for your positive comments

Reviewer 3 Report

Point 1: The authors need to mention the abbreviations such as LPS, GSDMA, NLRP3 and all the abbreviations must be elaborated when they first appear in the manuscript. A reader can not follow if the authors mention the abbreviations without elaborating them. For example, line 44, LPS (what it is? , please elaborate it). Similarly, for other words throughout the manuscript.

Response 1: Thanks for your comment. We have corrected the manuscript.

Point 2: It is not clear from the study design that why LPS was chosen for the study? Please provide a rationale.

Response 1: Thanks for your comment. Lipopolysaccharide (LPS) is a commonly used inflammatory booster that promotes inflammatory response and induces pyroptosis. We specify the intent to select LPS (The mice intraperitoneally injected with LPS were set as positive controls since it has been demonstrated that LPS may cause inflammation to accelerate pyroptosis). We have corrected the manuscript.

Point 3: In figure 5a, please include controls besides W24, W24 HCP1 and LPS. Discuss the results of 5a in the manuscript as well.

Response 1: Thanks for your comment. We have corrected the manuscript.

Point 4: In the title, the authors have mentioned that Hcp1 causes increased inflammation did not see any of this inflammation in the discussion section. Please provide reasonable rationale how this was observed.

Response 1: Thanks for your comment. We have corrected the manuscript. TNF-α, IFN-γ, IL-1β and IL-18 are typical inflammatory factors; a substantial release indicates a robust inflammatory response.

Thank the authors for the corrections and changes, however, all abbreviations in the manuscript must be described when they first mentioned in the manuscript. For example, KM mice, (need to include in the abstract what it stands for).

Author Response

(The authors gave the same response as above.)
